# Short-term and long-term revision rates after lumbar spine discectomy versus laminectomy: a population-based cohort study

Feng-Chen Kao,[1,2] Yao-Chun Hsu,[3,4,5,6,7] Chang-Bi Wang,[8] Yuan-Kun Tu,[1,2] Pao-Hsin Liu[9]

**Correspondence to**
Dr Yao-Chun Hsu;
holdenhsu@ntu.edu.tw

## ABSTRACT

**Background/objective** Degenerative diseases of the lumbar spine were managed with discectomy or laminectomy. This study aimed to compare these two surgical treatments in the postoperative revision rates.

**Design** A population-based cohort study from analysis of a healthcare database.

**Setting** Data were gathered from the Taiwan National Health Insurance Research Database (NHIRD).

**Participants** We enrolled 16 048 patients (4450 women and 11 598 men) with a mean age of 40.34 years who underwent lumbar discectomy or laminectomy for the first time between 1 January 1997 and 31 December 2007. All patients were followed up for 5 years or until death.

**Results** Revision rate within 3 months of the index surgery was significantly higher in patients who underwent discectomy (2.75%) than in those who underwent laminectomy (1.18%; p<0.0001). This difference persisted over the first year following the index surgery (3.38% vs 2.57%). One year afterwards, the revision rates were similar between the discectomy (9.75%) and laminectomy (9.69%) groups. The final spinal fusion surgery rates were also similar between the groups (11.25% vs 12.08%).

**Conclusion** The revision rate after lumbar discectomy was higher than that after laminectomy within 1 year of the index surgery. However, differences were not identified between patient groups for the two procedures with respect to long-term revision rates and the proportion of patients who required final spinal fusion surgery.

## INTRODUCTION

The natural progression of a degenerative spine leads to primary disc herniation and lumbar spinal stenosis, and most patients with these conditions are treated through surgical interventions.[1 2] Lumbar disc herniation is a common manifestation of degenerative lumbar disc disease[3–5] that occurs early in the degenerative cascade and involves tensile failure of the annulus to contain the gel-like nuclear portion of the disc. Although treatment for lumbar herniated discs can be challenging, non-surgical treatment is effective in most cases.[6 7] However, studies have

## Strengths and limitations of this study

► This population-based cohort study encompassed all residents of Taiwan.
► The universal and compulsory national health insurance mitigated attrition bias as no patients were lost to follow-up.
► However, radiographic and pathological data were unavailable in the Taiwan National Health Insurance Research Database. Therefore, we could not ascertain the level and pathology of the treated spine.
► The physical conditions of the patients could not be evaluated, and unmeasured confounding was possible.

indicated that surgery provides superior results to non-surgical treatments, especially with respect to short-term pain relief.[3 8]

Lumbar spinal stenosis is a progressive and dynamic disease that constitutes a continuum of pathological changes in the spinal column as a person ages. The likelihood of lumbar spinal stenosis increases during the fifth decade of life and ranges from 1.7% to 8% in the general population.[9] Surgical treatment focuses on a patient's pathological anatomy and involves relieving neurological compression; surgical procedures are usually more complex than those performed to relieve simple compression.[10]

Revision surgery, which is required in many cases of spinal disease after initial surgical treatment,[11 12] presents a challenge for spinal surgeons. Surgeons should be attuned to the clinical circumstances that are appropriate for additional surgery and should be technically qualified to address the anatomical and pathological obstacles involved in repeat surgery. Incidence of revision surgery after lumbar surgical discectomy varies from 0% to approximately 15%.[1] Frymoyer[13] reported incidence of postdiscectomy instability

requiring further spinal fusion surgery as high as 6.5%. Relatively few reports have specifically addressed revision surgery for lumbar spinal stenosis. Malter and colleagues[12] reported that the 5-year reoperation rate for patients with spinal stenosis was as high as 12%.

To investigate whether spinal reoperation rates differ after lumbar discectomy and laminectomy for lumbar spinal stenosis, we performed a population-based retrospective study of patients' 5-year follow-up data retrieved from the Taiwan National Health Insurance Research Database (NHIRD).

## DATA SOURCE

We examined data from the Taiwan NHIRD which is released by the Taiwan National Health Research Institute (NHRI) for public use. The NHRI covers the medical claims of 22.9 million residents of Taiwan, accounting for >99% of the total population. The NHIRD contains claims data from 1997 to 2013. The Department of Health and the National Health Insurance (NHI) Bureau of Taiwan ensure the completeness and accuracy of the NHIRD. This study was exempt from an ethics review because the medical records released by the insurance authority are encrypted secondary data and have been approved for use in research.

This retrospective population-based cohort study examined data from the longitudinal NHIRD. Until the end of 2013, all sampled individuals were followed up for outcome identification by using International Classification of Diseases, Ninth Revision, Clinical Modification (ICD-9-CM) codes.

## MATERIALS AND METHODS
### Patient and public involvement

Our study cohort included patients from the NHIRD who underwent lumbar discectomy or laminectomy for the first time between 1 January 1997 and 31 December 2007. Those who received their first lumbar discectomy or laminectomy after 2007 were excluded because dynamic stabilisation systems such as the Wallis system[14] were marketed in Taiwan after 2007. We also excluded individuals who were continually exposed to oral or injected forms of systemic corticosteroids for 6 months or longer, as well as those with diseases such as ankylosing spondylitis, systemic lupus erythematosus, rheumatoid arthritis, malignant cancers, spinal tumours, congenital spinal anomalies, spinal tuberculosis, spinal infections, spinal fractures, cervical spinal disease and thoracic spinal disease; the corresponding ICD-9-CM codes are listed in (online supplementary appendix 1).

We divided the study cohort into discectomy and laminectomy groups. Each patient's date of discharge from the hospital after their first lumbar discectomy or laminectomy was considered their index date. Revision lumbar spine surgery was defined as a second lumbar spine operation performed after the index date and comprised the following types: lumbar spine discectomy, lumbar spine laminectomy (including laminotomy) and lumbar spinal fusion surgery (with or without instrumentation). The revision rates in the two surgical groups were evaluated and compared, and the groups were propensity-score matched at a ratio of 1:1 based on the baseline characteristics of the patients. We assessed unmatched and matched data in this study.

Comorbidities existing prior to the index date were classified based on Charlson Comorbidity Index scores,[15] and incidences of mortality after the index dates were calculated for both groups. Mortality rates were considered when comparing revision rates to eliminate the influence of death on the calculated likelihood of revision surgery. We also calculated and compared the rates of final revision spinal fusion surgery in the two groups. All patients were followed up until death, withdrawal from the NHI programme or 31 December 2012.

## STATISTICAL ANALYSIS

We use Pearson's $\chi^2$ test and Yates's continuity correction to compare qualitative data, whereas the Student's t-test was employed for quantitative data. The annual revision rates were calculated with 95% CIs. The association between revision lumbar spine surgery between discectomy and laminectomy was explored by the Cox proportional hazard model that took into account age, gender and baseline comorbidity. Our study analysed the lumbar spine revision surgery rate by using the Fine and Gray regression model to calculate subdistribution hazards, and p values were determined using Gray's test. The subdistribution HR (sHR) was defined as significant when p<0.05. All statistical tests and calculations were performed using Statistical Analysis Software V.9.4 (SAS Institute).

## RESULTS
### Baseline characteristics of the patients

Our study cohort consisted of 66 754 patients (31 964 women and 34 790 men). The discectomy group comprised 27 867 patients and the laminectomy group comprised 38 887 patients. The unmatched and matched baseline characteristics and comorbidities of all patients are listed in table 1. After propensity-score matching, a total of 8024 patients were enrolled in this study. Lumbar spine revision surgery was defined as any of the following types of lumbar surgery performed after initial lumbar surgery: lumbar spine discectomy, lumbar spine laminectomy (including laminotomy) and lumbar spinal fusion surgery (with or without instrumentation). Final spinal fusion surgery referred to lumbar spinal fusion surgery (with or without instrumentation) performed during the follow-up period.

## REASONS OF LUMBAR SPINE REVISION SURGERY

Causes of lumbar spine revision surgeries are listed in online supplementary table S1.1 and S1.2. The prevalence

**Table 1** Characteristics and primary outcomes of patients who received laminectomy or discectomy surgeries

| 1. Unmatched baseline | Discectomy n=27 867 | Laminectomy n=38 887 | P values |
|---|---|---|---|
| Age | 47.83±15.58 | 59.91±14.02 | <0.0001 |
| Age group | | | <0.0001 |
| <20 | 416 (1.49) | 232 (0.60) | |
| 20–39 | 8987 (32.25) | 3667 (9.43) | |
| 40–59 | 11 511 (41.31) | 13 030 (33.51) | |
| 60–79 | 6663 (23.91) | 20 561 (52.87) | |
| ≥80 | 290 (1.04) | 1397 (3.59) | |
| Gender | | | <0.0001 |
| Female | 10 629 (38.14) | 21 335 (54.86) | |
| Male | 17 238 (61.86) | 17 552 (45.14) | |
| Comorbidities | | | |
| Myocardial infarct | 149 (0.53) | 404 (1.04) | <0.0001 |
| Congestive heart failure | 436 (1.56) | 1632 (4.20) | <0.0001 |
| Peripheral vascular disease | 196 (0.70) | 630 (1.62) | <0.0001 |
| Cerebrovascular disease | 1320 (4.74) | 4050 (10.41) | <0.0001 |
| Dementia | 199 (0.71) | 632 (1.63) | <0.0001 |
| Chronic lung disease | 514 (1.84) | 1620 (4.17) | <0.0001 |
| Connective tissue disease | 80 (0.29) | 132 (0.34) | 0.2357 |
| Ulcer | 5528 (19.84) | 11 362 (29.22) | <0.0001 |
| Chronic liver disease | 2593 (9.30) | 4768 (12.26) | <0.0001 |
| Diabetes | 2291 (8.22) | 5741 (14.76) | <0.0001 |
| Diabetes with end organ damage | 761 (2.73) | 2029 (5.22) | <0.0001 |
| Hemiplegia | 80 (0.29) | 238 (0.61) | <0.0001 |
| Moderate or severe kidney disease | 545 (1.96) | 1590 (4.09) | <0.0001 |
| Tumour, leukaemia, lymphoma | 20 (0.07) | 49 (0.13) | 0.0315 |
| Moderate or severe liver disease | 52 (0.19) | 98 (0.25) | 0.0784 |
| Malignant tumour, metastasis | | | – |
| AIDS | 4 (0.01) | 3 (0.01) | 0.4087 |
| Spinal revision surgery (3 month) | 765 (2.75) | 459 (1.18) | <0.0001 |
| Discectomy | 449 (1.61) | 128 (0.33) | <0.0001 |
| Laminectomy | 187 (0.67) | 196 (0.5) | 0.0048 |
| Spinal instrumentation | 129 (0.46) | 135 (0.35) | 0.0188 |
| Spinal revision surgery (3 month~1 year) | 941 (3.38) | 999 (2.57) | <0.0001 |
| Discectomy | 389 (1.40) | 186 (0.48) | <0.0001 |
| Laminectomy | 287 (1.03) | 406 (1.04) | 0.8587 |
| Spinal instrumentation | 265 (0.95) | 407 (1.05) | 0.2220 |
| Spinal revision surgery (>1 year) | 2718 (9.75) | 3770 (9.69) | 0.8006 |
| Discectomy | 844 (3.03) | 485 (1.25) | <0.0001 |
| Laminectomy | 708 (2.54) | 1282 (3.3) | <0.0001 |
| Spinal instrumentation | 1166 (4.18) | 2003 (5.15) | <0.0001 |
| Total spinal revision surgery | 4424 (15.88) | 5228 (13.44) | <0.0001 |
| Discectomy | 1682 (6.04) | 799 (2.05) | <0.0001 |
| Laminectomy | 1182 (4.24) | 1884 (4.84) | 0.0002 |
| Spinal instrumentation | 1560 (5.60) | 2545 (6.54) | <0.0001 |

Continued

**Table 1**  Continued

| 1. Unmatched baseline | Discectomy n=27 867 | Laminectomy n=38 887 | P values |
|---|---|---|---|
| Final spinal fusion | 3136 (11.25) | 4699 (12.08) | 0.0010 |
| Death | 3900 (14.00) | 8545 (21.97) | <0.0001 |

| 2. Matched baseline | Discectomy n=8024 | Laminectomy n=8024 | P values |
|---|---|---|---|
| Age | 40.16±11.26 | 40.51±11.51 | 0.0536 |
| Age group | | | 0.3398 |
| <20 | 195 (2.43) | 217 (2.70) | |
| 20–39 | 3621 (45.13) | 3500 (43.62) | |
| 40–59 | 3922 (48.88) | 4023 (50.14) | |
| 60–79 | 246 (3.07) | 244 (3.04) | |
| ≥80 | 40 (0.50) | 40 (0.50) | |
| Gender | | | 1.0000 |
| Female | 2225 (27.73) | 2225 (27.73) | |
| Male | 5799 (72.27) | 5799 (72.27) | |
| Comorbidities | | | |
| Myocardial infarct | 32 (0.40) | 34 (0.42) | 0.8051 |
| Congestive heart failure | 87 (1.08) | 88 (1.10) | 0.9394 |
| Peripheral vascular disease | 49 (0.61) | 60 (0.75) | 0.2904 |
| Cerebrovascular disease | 215 (2.68) | 220 (2.74) | 0.8080 |
| Dementia | 41 (0.51) | 44 (0.55) | 0.7442 |
| Chronic lung disease | 86 (1.07) | 79 (0.98) | 0.5838 |
| Connective tissue disease | 15 (0.19) | 17 (0.21) | 0.7234 |
| Ulcer | 1124 (14.01) | 1129 (14.07) | 0.9095 |
| Chronic liver disease | 705 (8.79) | 693 (8.64) | 0.7369 |
| Diabetes | 431 (5.37) | 412 (5.13) | 0.5014 |
| Diabetes with end organ damage | 150 (1.87) | 144 (1.79) | 0.7240 |
| Hemiplegia | 18 (0.22) | 17 (0.21) | 0.8656 |
| Moderate or severe kidney disease | 107 (1.33) | 113 (1.41) | 0.6838 |
| Tumour, leukaemia, lymphoma | 3 (0.04) | 4 (0.05) | 0.7054 |
| Moderate or severe liver disease | 7 (0.09) | 10 (0.12) | 0.4666 |
| Malignant tumour, metastasis | | | – |
| AIDS | | | – |
| Spinal revision surgery (3 month) | 208 (2.59) | 123 (1.53) | <0.0001 |
| Discectomy | 128 (1.60) | 48 (0.60) | <0.0001 |
| Laminectomy | 46 (0.57) | 37 (0.46) | 0.3220 |
| Spinal instrumentation | 34 (0.42) | 38 (0.47) | 0.6366 |
| Spinal revision surgery (3 month~1 year) | 241 (3.00) | 189 (2.36) | 0.0110 |
| Discectomy | 109 (1.36) | 54 (0.67) | <0.0001 |
| Laminectomy | 58 (0.72) | 63 (0.79) | 0.6482 |
| Spinal instrumentation | 74 (0.92) | 72 (0.90) | 0.8679 |
| Spinal revision surgery (>1 year) | 675 (8.41) | 665 (8.29) | 0.7754 |
| Discectomy | 278 (3.46) | 181 (2.26) | <0.0001 |
| Laminectomy | 132 (1.65) | 164 (2.04) | 0.0605 |
| Spinal instrumentation | 265 (3.30) | 320 (3.99) | 0.0205 |

Continued

| Table 1 Continued | | | |
| --- | --- | --- | --- |
| 2. Matched baseline | Discectomy n=8024 | Laminectomy n=8024 | P values |
| Total spinal revision surgery | 1124 (14.01) | 977 (12.18) | 0.0006 |
| Discectomy | 515 (6.42) | 283 (3.53) | <0.0001 |
| Laminectomy | 236 (2.94) | 264 (3.29) | 0.2033 |
| Spinal instrumentation | 373 (4.65) | 430 (5.36) | 0.0390 |
| Final spinal fusion | 784 (9.77) | 838 (10.44) | 0.1573 |
| Death | 795 (9.91) | 884 (11.02) | 0.0217 |

of incidental durotomy was 0.04%. The proportions of postoperative haemorrhage and postoperative spine infection were 0.18% and 1.73%, respectively. Finally, the lumbar disc pathology rate was 40.74%.

## TOTAL SPINAL SURGERY REVISION RATES

The annual revision rates in the discectomy and laminectomy groups were 5.63% (95% CI 5.15% to 6.16%) and 3.92% (95% CI 3.52% to 4.37%), respectively. Values representing cumulative incidence of revision spinal surgery are displayed in figure 1. Significant differences in total revision spinal surgery rates between patients

who received lumbar discectomy and those who received lumbar laminectomy as initial surgery were identified. In the unmatched data, the revision spinal surgery rates in the discectomy and laminectomy groups were 15.88% and 13.44%, respectively (p<0.0001). In the matched data, the corresponding rates were 14.01% and 12.18%, respectively (p<0.001).

## RATES FOR REVISION SURGERY PERFORMED WITHIN 3 MONTHS OF INITIAL SPINAL SURGERY

The rates for revision spinal surgery performed within 3 months of initial spinal surgery significantly differed

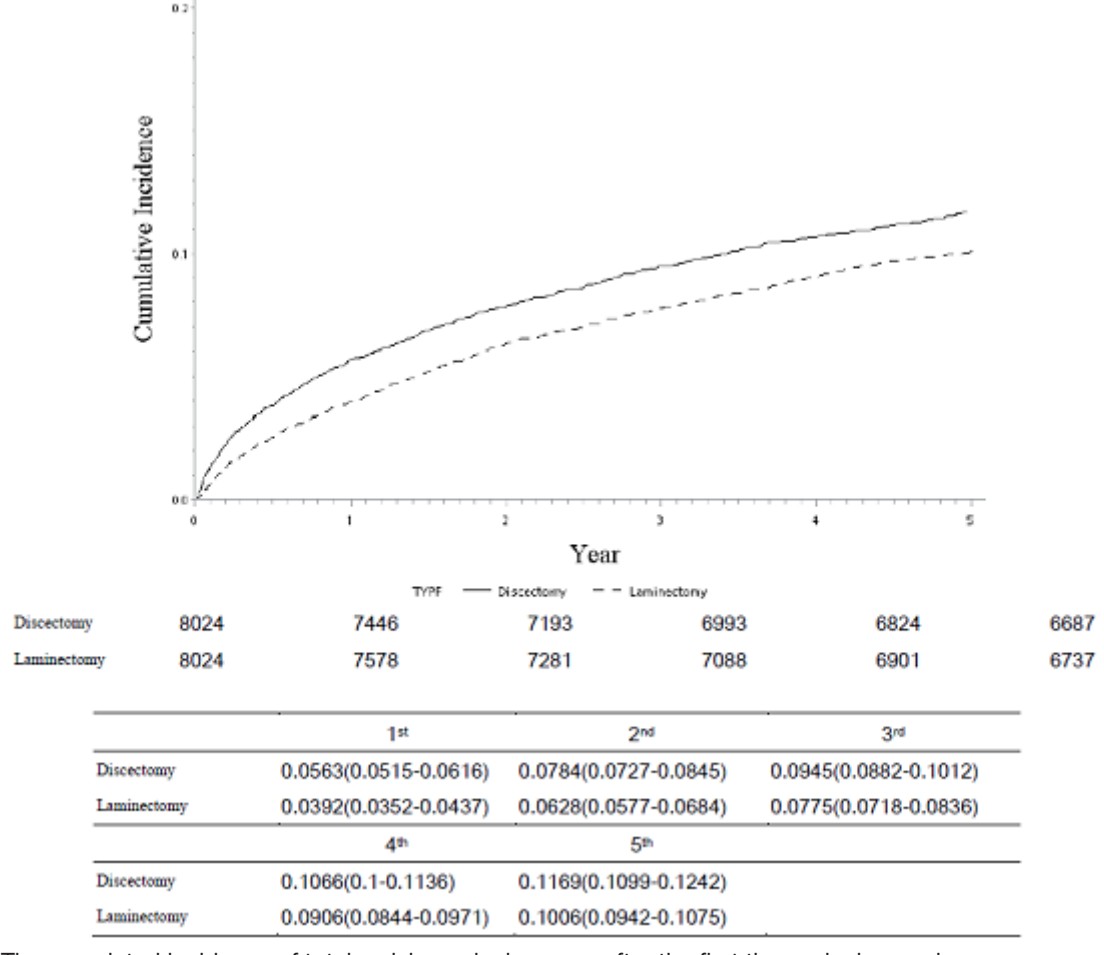

| | 1st | 2nd | 3rd |
| --- | --- | --- | --- |
| Discectomy | 0.0563(0.0515-0.0616) | 0.0784(0.0727-0.0845) | 0.0945(0.0882-0.1012) |
| Laminectomy | 0.0392(0.0352-0.0437) | 0.0628(0.0577-0.0684) | 0.0775(0.0718-0.0836) |
| | 4th | 5th | |
| Discectomy | 0.1066(0.1-0.1136) | 0.1169(0.1099-0.1242) | |
| Laminectomy | 0.0906(0.0844-0.0971) | 0.1006(0.0942-0.1075) | |

**Figure 1** The cumulated incidence of total revision spinal surgery after the first time spinal surgeries.

between the two groups (p<0.0001). Based on the unmatched data, the revision spinal surgery rates in the discectomy and laminectomy groups were 2.75% and 1.18%, respectively. In the matched data, the corresponding rates were 2.59% and 1.53%, respectively.

## RATES FOR REVISION SURGERY PERFORMED BETWEEN 3 MONTHS AND 1 YEAR AFTER INITIAL SPINAL SURGERY

The rates for revision spinal surgery performed between 3 months and 1 year after initial spinal surgery also significantly differed between patients who initially received lumbar discectomy and those who initially received lumbar laminectomy. In the unmatched data, the revision spinal surgery rates in the discectomy and laminectomy groups were 3.38% and 2.57%, respectively (p<0.0001). In the matched data, the corresponding rates were 3.00% and 2.36%, respectively (p<0.05).

## RATES FOR REVISION SURGERY PERFORMED MORE THAN 1 YEAR AFTER INITIAL SPINAL SURGERY

The rates for revision spinal surgery performed more than 1 year after initial spinal surgery did not significantly differ between patients who initially received

lumbar discectomy and those who initially received lumbar laminectomy. In the unmatched data, the revision spinal surgery rates in the discectomy and laminectomy groups were 9.75% and 9.69%, respectively. In the matched data, the corresponding rates were 8.41% and 8.29%, respectively.

## DIFFERENCES IN MULTIVARIATE-ADJUSTED TOTAL REVISION SPINAL SURGERY RATES BETWEEN DISCECTOMY AND LAMINECTOMY GROUPS

A multivariate-adjusted Cox proportional hazards model revealed independent differences in the unmatched and matched data (adjusted sHRs 0.81 and 0.86, respectively; 95% CIs 0.78 to 0.85 and 0.79 to 0.94, respectively; table 2) between the discectomy and laminectomy groups. Analysis of the unmatched data (table 2) revealed that age (sHR 1.01; 95% CI 1.00 to 1.01), sex (sHR 1.09; 95% CI 1.05 to 1.14), peripheral vascular disease (sHR 0.73; 95% CI 0.59 to 0.91) and diabetes mellitus (DM; sHR 1.09; 95% CI 1.01 to 1.17) were the risk factors responsible for differences in spinal revision rates between the discectomy and laminectomy groups. Analysis of the matched data indicated that age (sHR

Table 2  Multivariate Cox proportional hazard models for revision lumbar spine surgical rates between discectomy and laminectomy with or without matched data

| | Unmatched | | Matched | |
|---|---|---|---|---|
| | sHR (95% CI) | P values | sHR (95% CI) | P values |
| Laminectomy vs discectomy | 0.81 (0.78 to 0.85) | <0.0001 | 0.86 (0.79 to 0.94) | 0.0007 |
| Age | 1.01 (1.00 to 1.01) | <0.0001 | 1.01 (1.00 to 1.01) | 0.0007 |
| Male vs female | 1.09 (1.05 to 1.14) | <0.0001 | 1.09 (0.99 to 1.20) | 0.0937 |
| Comorbidities | | | | |
| Myocardial infarct | 1.15 (0.94 to 1.42) | 0.1825 | 1.21 (0.69 to 2.14) | 0.5097 |
| Congestive heart failure | 1.04 (0.92 to 1.18) | 0.4979 | 1.30 (0.89 to 1.90) | 0.1751 |
| Peripheral vascular disease | 0.73 (0.59 to 0.91) | 0.0046 | 0.82 (0.48 to 1.41) | 0.4788 |
| Cerebrovascular disease | 0.99 (0.91 to 1.07) | 0.7413 | 0.97 (0.73 to 1.28) | 0.8136 |
| Dementia | 1.11 (0.92 to 1.33) | 0.2813 | 1.19 (0.69 to 2.05) | 0.5300 |
| Chronic lung disease | 1.05 (0.94 to 1.18) | 0.4067 | 1.00 (0.67 to 1.50) | 0.9833 |
| Connective tissue disease | 1.16 (0.83 to 1.61) | 0.3925 | 1.73 (0.86 to 3.49) | 0.1262 |
| Ulcer | 0.96 (0.92 to 1.01) | 0.1285 | 1.12 (0.99 to 1.27) | 0.0854 |
| Chronic liver disease | 0.99 (0.92 to 1.06) | 0.7486 | 1.14 (0.98 to 1.33) | 0.0917 |
| Diabetes | 1.09 (1.01 to 1.17) | 0.0263 | 1.14 (0.92 to 1.42) | 0.2392 |
| Diabetes with end organ damage | 1.12 (1.00 to 1.25) | 0.0590 | 0.99 (0.70 to 1.40) | 0.9436 |
| Hemiplegia | 1.18 (0.88 to 1.57) | 0.2672 | 1.18 (0.52 to 2.72) | 0.6897 |
| Moderate or severe kidney disease | 1.09 (0.97 to 1.23) | 0.1319 | 0.83 (0.57 to 1.22) | 0.3431 |
| Tumour, leukaemia, lymphoma | 1.40 (0.80 to 2.47) | 0.2434 | NA | |
| Moderate or severe liver disease | 1.36 (0.90 to 2.06) | 0.1399 | 1.34 (0.43 to 4.22) | 0.6124 |
| Malignant tumour, metastasis | NA | | NA | – |
| AIDS | 1.11 (0.16 to 7.90) | 0.9149 | NA | – |

sHR, subdistribution HR.

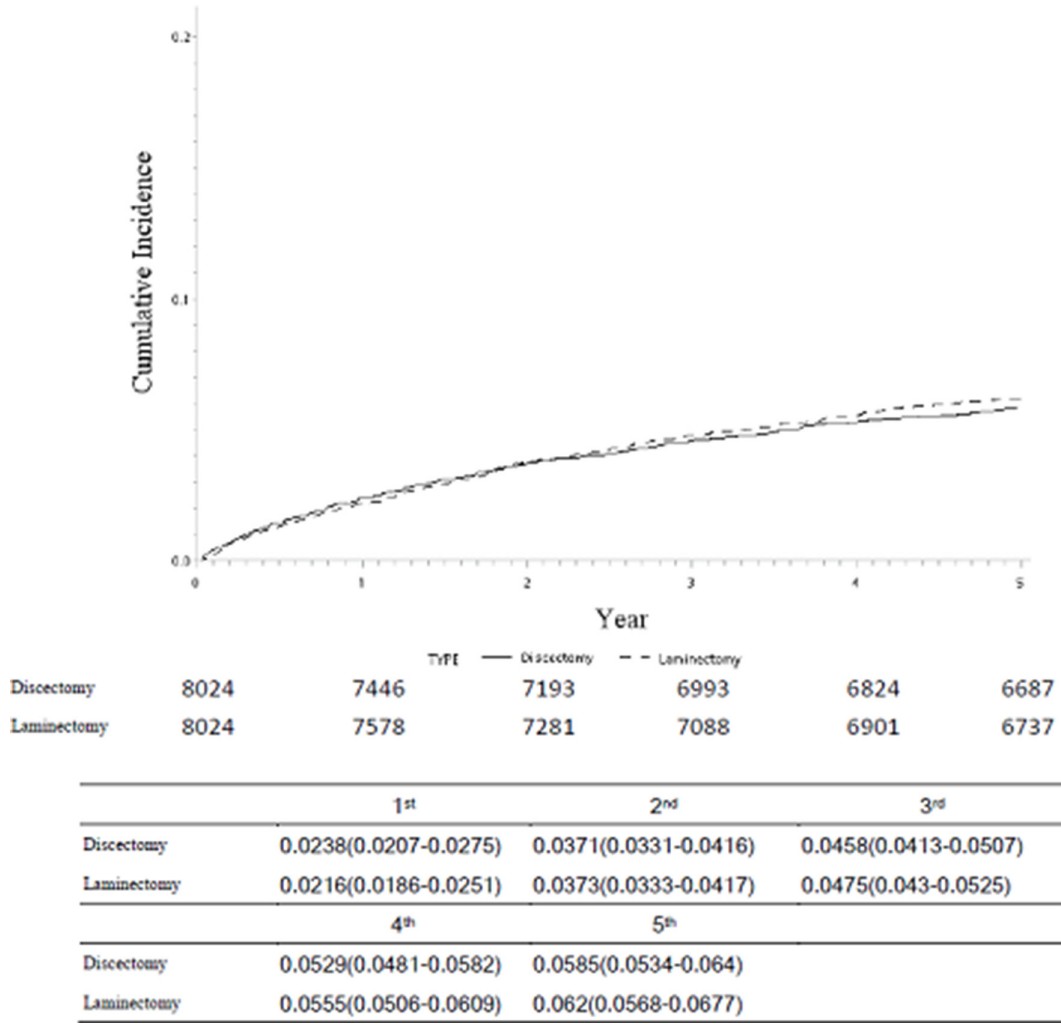

| | 1st | 2nd | 3rd |
|---|---|---|---|
| Discectomy | 0.0238(0.0207-0.0275) | 0.0371(0.0331-0.0416) | 0.0458(0.0413-0.0507) |
| Laminectomy | 0.0216(0.0186-0.0251) | 0.0373(0.0333-0.0417) | 0.0475(0.043-0.0525) |
| | 4th | 5th | |
| Discectomy | 0.0529(0.0481-0.0582) | 0.0585(0.0534-0.064) | |
| Laminectomy | 0.0555(0.0506-0.0609) | 0.062(0.0568-0.0677) | |

**Figure 2** The cumulated incidence of final spinal fusion surgery after the first time spinal surgeries.

1.01; 95% CI 1.00 to 1.01) was the risk factor responsible for differences in spinal revision rates between the two groups.

## RATES FOR FINAL SPINAL FUSION SURGERY PERFORMED AFTER INITIAL SPINAL SURGERY

The annual revision rates in the discectomy and laminectomy groups were 2.38% (95% CI 2.07% to 2.75%) and 2.16% (95% CI 1.86% to 2.51%), respectively. The value representing cumulative incidence of final spinal fusion surgery performed after initial spinal surgery is displayed in figure 2. No significant differences in the rates for final spinal fusion surgery performed after initial surgery were identified between patients who initially received lumbar discectomy and those who initially received lumbar laminectomy. In the unmatched data, the final spinal fusion surgery rates in the discectomy and laminectomy groups were 11.25% and 12.08%, respectively. In the matched data, the corresponding rates were 9.77% and 10.44%, respectively.

## DIFFERENCES IN MULTIVARIATE-ADJUSTED RATES OF FINAL SPINAL FUSION SURGERY PERFORMED AFTER INITIAL SPINAL SURGERY BETWEEN DISCECTOMY AND LAMINECTOMY GROUPS

The multivariate-adjusted Cox proportional hazards model revealed no differences in the unmatched data between the discectomy and laminectomy groups (adjusted sHR 1.05; 95% CI 1.00 to 1.10; table 3). However, the model revealed independent differences in the matched data between the groups (adjusted sHR 1.11; 95% CI 1.01 to 1.22). In the unmatched data analysis (table 3), age (sHR 1.00; 95% CI 1.00 to 1.01), chronic lung disease (sHR 1.15; 95% CI 1.01 to 1.30), ulcer (sHR 1.18; 95% CI 1.12 to 1.24), chronic liver disease (sHR 1.21; 95% CI 1.13 to 1.30), DM (sHR 1.29; 95% CI 1.19 to 1.39) and moderate or severe kidney disease (sHR 1.20; 95% CI 1.06 to 1.36) were the risk factors for different final spinal fusion rates between the discectomy and laminectomy groups. In the matched data analysis, age (sHR 1.02; 95% CI 1.01 to 1.02), ulcer (sHR 1.34; 95% CI 1.16 to 1.55) and chronic liver disease (sHR 1.37; 95% CI 1.16 to 1.62) were the corresponding risk factors.

**Table 3** Multivariate Cox proportional hazard models for final revision lumbar spine fusion rates between discectomy and laminectomy with or without matched data

| | Unmatched | | Matched | |
| --- | --- | --- | --- | --- |
| | sHR (95% CI) | P values | sHR (95% CI) | P values |
| Laminectomy vs discectomy | 1.05 (1.00 to 1.10) | 0.0524 | 1.11 (1.01 to 1.22) | 0.0377 |
| Age | 1.00 (1.00 to 1.01) | <0.0001 | 1.02 (1.01 to 1.02) | <0.0001 |
| Comorbidities | | | | |
| Myocardial infarct | 1.16 (0.92 to 1.45) | 0.2131 | 0.95 (0.47 to 1.91) | 0.8832 |
| Congestive heart failure | 1.06 (0.93 to 1.21) | 0.4071 | 1.16 (0.75 to 1.78) | 0.5045 |
| Peripheral vascular disease | 0.96 (0.78 to 1.18) | 0.6927 | 0.90 (0.52 to 1.57) | 0.7183 |
| Cerebrovascular disease | 1.04 (0.95 to 1.13) | 0.3858 | 1.07 (0.80 to 1.45) | 0.6419 |
| Dementia | 1.13 (0.93 to 1.38) | 0.2320 | 0.87 (0.45 to 1.69) | 0.6863 |
| Chronic lung disease | 1.15 (1.01 to 1.30) | 0.0351 | 0.95 (0.61 to 1.50) | 0.8351 |
| Connective tissue disease | 0.89 (0.59 to 1.34) | 0.5653 | 1.09 (0.46 to 2.60) | 0.8492 |
| Ulcer | 1.18 (1.12 to 1.24) | <0.0001 | 1.34 (1.16 to 1.55) | <0.0001 |
| Chronic liver disease | 1.21 (1.13 to 1.30) | <0.0001 | 1.37 (1.16 to 1.62) | 0.0002 |
| Diabetes | 1.29 (1.19 to 1.39) | <0.0001 | 1.19 (0.93 to 1.54) | 0.1730 |
| Diabetes with end organ damage | 1.11 (0.98 to 1.25) | 0.0887 | 0.95 (0.65 to 1.40) | 0.7954 |
| Hemiplegia | 1.12 (0.80 to 1.56) | 0.5194 | 0.43 (0.10 to 1.80) | 0.2471 |
| Moderate or severe kidney disease | 1.20 (1.06 to 1.36) | 0.0042 | 1.04 (0.71 to 1.53) | 0.8466 |
| Tumour, leukaemia, lymphoma | 1.31 (0.71 to 2.41) | 0.3819 | NA | |
| Moderate or severe liver disease | 1.36 (0.87 to 2.13) | 0.1778 | 1.02 (0.26 to 4.01) | 0.9728 |
| Malignant tumour, metastasis | NA | | NA | – |
| AIDS | 1.91 (0.32 to 11.39) | 0.4762 | NA | – |

sHR, subdistribution HR.

## DISCUSSION

Lumbar disc herniation is one of the most common lumbar spine disorders.[16] In 1934, Mixter and Barr[17] identified a link between sciatica and lumbar disc herniation; since this discovery, discectomy through limited laminotomy has been the most common form of surgical management for lumbar disc prolapse in cases of conservative management failure.[18] The efficacy of lumbar discectomy for treating lumbar disc herniation has been demonstrated[19 20]; however, unsatisfactory outcomes after lumbar discectomy have been reported in 5%–20% of cases.[21–24] The Spine Patient Outcomes Research Trial reported that in patients with lumbar disc herniation, the proportions of reoperation within 4 and 8 years of index procedures were as high as 9% for discectomy patients and 13% for laminectomy patients.[19] The most common cause of ongoing disability after lumbar discectomy is recurrent lumbar disc herniation which occurs in 5%–15% of patients (this incidence proportion increases over time).[21 23 25–28] In our study cohort, the rates for revision spinal surgery performed within 3 months and 1 year of lumbar discectomy were 2.75% and 3.38%, respectively; those for revision surgery performed after 1 year and of total revision surgery were 9.75% and 15.88%, respectively.

Lumbar stenosis is caused by spondylotic changes in the facet joints, spinal instability or a congenitally small spinal canal.[29] Laminectomy remains the standard treatment for spinal stenosis when the spine does not exhibit instability.[29] Despite adequate lumbar decompression, substantial postoperative back and leg pain occur in 10%–15% of patients.[30] Historically, a high proportion of lumbar laminectomies fail, and the proportion of patients who experience recurrent back pain may reach 47%.[31 32] No reoperation rates after lumbar laminectomy without spinal fusion surgery have been reported. In our study, the rates for revision spinal surgery performed within 3 months and 1 year of lumbar laminectomy were 1.18% and 2.57%, respectively; those for revision surgery performed after 1 year and for total revision surgery were 9.69% and 13.44%, respectively.

Spinal structures that contribute to spinal stability in certain proportions of patients are as follows: facet capsule 39%, disc and annulus 29%, supraspinous and intraspinous ligaments 19% and ligamentum flavum 13%.[33] Interventions at the hemilamina and ligamentum flavum can change both the load-bearing and kinematic characteristics of the spine and lead to spinal segment hypermobility and accelerated bone degeneration.[34 35] Even microdiscectomy can increase the risk of single-level

instability.[36] Extensive laminectomy can also potentiate spinal instability.[37 38] Lai *et al*[39] reported that sacrificing supraspinous ligaments or tendon insertion points in spinous processes can accelerate development of adjacent instability. Incidences of adjacent instability increase with the number of destructed laminae, and far more posterior spinal complexes are destructed in lumbar laminectomy than in lumbar discectomy. Hence, theoretically, lumbar laminectomy causes greater spinal instability than does lumbar discectomy, leading to a higher reoperation rate after lumbar laminectomy.

In contrast to the theoretically expected outcomes, our study revealed independent differences in reoperation rates based on the unmatched and matched data (adjusted sHR 0.81 and 0.86; 95% CI 0.78 to 0.85 and 0.79 to 0.94, respectively) between the discectomy and laminectomy groups. Based on the unmatched data, revision spinal surgery rates in the discectomy and laminectomy groups were 15.88% and 13.44%, respectively (p<0.0001). According to the matched data, the corresponding rates in the discectomy and laminectomy groups were 14.01% and 12.18%, respectively (p<0.001). Compared with the laminectomy group, the discectomy group had higher rates of reoperation within 3 months and between 3 months and 1 year after initial surgery (p<0.05). However, beyond 1 year, the reoperation rates did not significantly differ between the laminectomy and discectomy groups.

Numerous reasons for reoperation after discectomy have been suggested. Early recurrence may be due to reherniation, infection or arachnoiditis, whereas late recurrence may be attributed to foraminal stenosis, a painful disc, epidural fibrosis, iatrogenic segmental instability, progressive facet degeneration or sacroiliac joint pain.[40–42] Outcomes based on natural degeneration of the lumbar spine more than 1 year after initial lumbar spine surgery were similar in the discectomy and laminectomy groups.

North *et al*[43] reported that incidence of instability increased from 12.5% after initial revision surgery to 50% after the fourth surgery. Fusion of the symptomatic spinal segment during revision spinal surgery is related to successful outcomes.[44–47] In our study, no significant differences were observed in the final spinal fusion surgery rates after initial spinal surgery between patients who received lumbar discectomy (11.25%) and those who received lumbar laminectomy (12.08%).

Our study had some limitations. First, the laboratory, radiographic and pathological data of the patients were unavailable in the NHIRD. Thus, we were unable to differentiate between true lumbar disc prolapse and spinal canal stenosis. Second, the physical conditions of the study cohort patients could not be evaluated; this may have led to healthy patient bias. Nevertheless, this stringent definition would have biased the results towards a null association rather than creating a spurious one. In addition, the potential influence of body weight, habitual cigarette smoking, alcohol consumption and dietary habits could not be assessed because related information

was unavailable in the NHIRD. We were also unable to acquire direct information on these factors because linking the NHIRD with external databases is strictly prohibited for privacy protection. However, an advantage of the NHIRD is its inclusion of information on 99% of the residents of Taiwan, and no patients in our NHIRD study cohort were lost to follow-up. The complete follow-up in this study was particularly attributable to hospital accessibility.

In conclusion, rates for reoperation within 1 year were higher after lumbar discectomy than after lumbar laminectomy. Beyond 1 year after initial lumbar surgery, reoperation rates and final lumbar spinal fusion surgery rates were similar in the discectomy and laminectomy groups.

**Author affiliations**
[1]Department of Orthopedics, E-Da Hospital, Kaohsiung, Taiwan
[2]School of Medicine for International Students, I-Shou University, Kaohsiung, Taiwan
[3]School of Medicine, Fu-Jen Catholic University, New Taipei, Taiwan
[4]Big Data Research Center, Fu-Jen Catholic University, New Taipei, Taiwan
[5]Division of Gastroenterology, Fu-Jen Catholic University Hospital, New Taipei, Taiwan
[6]Graduate Institute of Clinical Medicine, China Medical University, Taichung, Taiwan
[7]Division of Gastroenterology and Hepatology, E-Da Hospital, Kaohsiung, Taiwan
[8]Graduate Institute of Public Health, China Medical University, Taichung, Taiwan
[9]Department of Biomedical Engineering, I-Shou University, Kaohsiung, Taiwan

**Contributors** Substantial contributions to the conception or design of this work or the acquisition, analysis, or interpretation of data for this work: Y-CH and Y-KT. Drafting the work or revising it critically for important intellectual content: F-CK, C-BW and P-HL. Final approval of the version to be published: C-BW and P-HL. Agreement to be accountable for all aspects of this work in ensuring that questions related to the accuracy or integrity of any part of this work are appropriately investigated and resolved: Y-CH and Y-KT.

**Funding** This work was supported by the Center for Database Research, E-DA HEALTHCARE GROUP, and E-Da Hospital (EDAHP103048).

**Competing interests** None declared.

**Patient consent** Not required.

**Ethics approval** Institutional Review Board of E-Da hospital (EMRP-104–04) and the Taiwan National Health Research Institute (NHIRD-104–167).

**Provenance and peer review** Not commissioned; externally peer reviewed.

**Data sharing statement** No additional data available.

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
