## [Reviewer comments · BMJ Open]

ARTICLE DETAILS

TITLE (PROVISIONAL)	Short-term and Long-term Revision Rates after Lumbar Spine Discectomy versus Laminectomy: A Population-Based Cohort Study
AUTHORS	kao, Feng-Chen; Hsu, Yao-chun; Wang, Chang-Bi; Tu, Yuan-Kun; Liu, Pao-Hsin

VERSION 1 – REVIEW

REVIEWER	DR.G.SUDHIR Sri Ramachandra Medical University, Tamil Nadu, India
REVIEW RETURNED	28-Jan-2018

GENERAL COMMENTS	Authors have not given a clear description about the causes for revision surgeries. Laminectomy involves much more destabilising potential than discectomy. In that case what is the reason for the results obtained? If the reasons for revision surgery were explained clearly, then it would have been useful.
---

REVIEWER	Hazem Elsebaie Cairo University, Egypt
REVIEW RETURNED	29-Jan-2018

GENERAL COMMENTS	I believe the manuscript lacks many important details that can make it beneficial to the reader; otherwise it would be just statistical analysis without clinical relevance nor valuable conclusion especially that the authors are discussing 2 different surgeries for 2 different pathologies. To make it of value the authors need to clarify many pre and postoperative data. 1. Lumbar disc prolapse and spinal canal stenosis are in many times happening together. What was the pathology in the 2nd group to perform only laminectomy for all these patients? Was it a pure bony stenosis in all these patients or was it surgeons preference to do a laminectomy for lumbar disc prolapse? It is important to know the primary pathology of these patients. Most of the time the surgery itself is a mixture between discectomy and decompression. Did the authors check that none of the discectomy patients underwent laminectomy as well? And if so how many? 2. It is also important to know the cause of revision: was it at the same level? Was it recurrent of the same pathology or different pathology? Was the cause of revision neurological or mechanical or both? Was it due to infection?
---

REVIEWER	Shahnaz KLOUCHE, MD Clinique du Sport Paris, FRANCE
REVIEW RETURNED	01-Mar-2018

GENERAL COMMENTS	The study is interesting but the manuscript needs some clarifications and the statistics need to be more explicated and completed.  1. In the abstract:  a. the main evaluation criterion is unclear b. there is no information about patients (number, mean age, sex ratio...). Please refer to STROBE guidelines. 2. Methods  a. Please present the main and secondary criteria in a separate paragraph b. In statistical analysis paragraph, please note that survival analysis was performed. Did you calculate the annual revision rate with 95%CI ? This is a useful information to provide to reader. c. You used some baseline characteristics to match you patients in ratio 1:1. Are these variables adapted for the event "revision of lumbar spinal surgery" or "non fusion"? d. What are the risk factors for revision and non-fusion? Why not consider these factors in your multivariate analysis? e. You used the Logrank for which variables? 3. Results  a. In the description of patients, there is no information regarding the surgical technique and possible devices used. b. Please add annual revision rate and annual fusion rate with 95%CI c. Do you have the rea
--

VERSION 1 – AUTHOR RESPONSE

The reviewer's comments

Editorial Requests:

1. Please revise your title so that it includes your study's setting. This is the preferred format for the journal. It should also be clearer what the research question is.

R: Thank you for the suggestion. We have revise our title as " Short-term and Long-term Revision Rates after Lumbar Spine Discectomy versus Laminectomy: A Population-Based Cohort Study ". (line1-2, page1, in the Title)

2. Please improve your abstract. For example, the 'Objectives' section does not include any objectives. Please also revise the "Patients or participants" sub-heading to "Participants" and include the sample size.).

R: Thank you for the suggestion, according to which we rewrite the abstract to clarify the 'Objectives' and "Participants" section (line 29-36, page2, in the Abstract).

3. The quality of English is not at the requisite standard for publication. We strongly recommend consulting a native English speaker or professional copy-editing service. You need to pay particular attention to the abstract and the strengths and limitations section.

R: We are indebted to you for helping us to improve our manuscript. We have sent the manuscript to a company to edit the English.

4. The “article summary” section after the abstract needs to be changed to “strengths and limitations” and re-written. Each bullet point should be a specific strength or limitation of your study relating to the study's design or methods. It should not be a summary of the study and its findings.

R: Thank you for reminding us to comply with the journal style. We have revised the “strengths and limitations” accordingly. (line 50-58, page 3, in the Abstract).

5. The contributorship statement on page 15 needs improving. Please re-write this section so that it aligns with the ICMJE criteria for authorship.

R: We re-write this section according to the ICMJE criteria. (line 361-369, page 16-17, in the Contribution statement)

Reviewer 1

1. Authors have not given a clear description about the causes for revision surgeries. Laminectomy involves much more destabilising potential than discectomy. In that case what is the reason for the results obtained? If the reasons for revision surgery were explained clearly, then it would have been useful.

R: Thank you for reminding us to clarify this important point. We gather the information of the reason for revision surgeries from NHIRD. We calculated the result and add a section of "Reasons of Lumbar Spine Revision Surgery" and table S1.1 and table S1.2 (line 179-183, page9, in the Results).

Reviewer 2

1. 1. Lumbar disc prolapse and spinal canal stenosis are in many times happening together. What was the pathology in the 2nd group to perform only laminectomy for all these patients? Was it a pure bony stenosis in all these patients or was it surgeons preference to do a laminectomy for lumbar disc prolapse? It is important to know the primary pathology of these patients. Most of the time the surgery itself is a mixture between discectomy and decompression. Did the authors check that none of the discectomy patients underwent laminectomy as well? And if so how many?.

R: Thank you for raising this important point that needed clarification. Indeed, lumbar disc prolapse and spinal canal stenosis could coexist and it was possible that the surgery could be a mixture between discectomy and decompression. In the cases with a mixed operation, the major surgery would be coded for reimbursement according to the insurance policy. Therefore, even if some patients received discectomy plus decompression, they were still grouped by the predominant surgery. Regrettably, we could not confirm how many surgical procedures were mixed. We have add the point in our "strengths and limitations" part. (line 55-56, page 3, in the Abstract).

2. It is also important to know the cause of revision: was it at the same level ? Was it recurrent of the same pathology or different pathology? Was the cause of revision neurological or mechanical or both? Was it due to infection?

R: We agree with you that the cause of revision should have been considered for a better interpretation of our data. We gather the information of the reason for revision surgeries from NHIRD. We calculated the result and add a section of " Reasons of Lumbar Spine Revision Surgery" and table S1.1 and table S1.2 (line 179-183, page 9, in the Results).

Reviewer 3

1. In the abstract:

a. the main evaluation criterion is unclear

b. there is no information about patients (number, mean age, sex ratio...). Please refer to STROBE guidelines.

R: Thank you for reminding us to adhere to the STROBE guidelines. We have revised the abstract to clarify the main evaluation criterion and patient's information (line 29-36, page1, in the Abstract).

2. Methods

a. Please present the main and secondary criteria in a separate paragraph

b. In statistical analysis paragraph, please note that survival analysis was performed. Did you calculate the annual revision rate with 95%CI ? This is a useful information to provide to reader.

- c. You used some baseline characteristics to match you patients in ratio 1:1. Are these variables adapted for the event "revision of lumbar spinal surgery" or "non fusion"?
- d. What are the risk factors for revision and non-fusion? Why not consider these factors in your multivariate analysis?
- e. You used the Logrank for which variables?

R: Thank you for your careful review and constructive comments.

- a. We separate the main and secondary criteria in a separate paragraph. (line 146-152, page 7-8, in the Methods)
- b. Yes, we used survival analysis. We presented the annual revision rate with 95% CI (line 155-163, page8, in the Methods)
- c. Age and sex were risk factors for the revision of lumbar spinal surgery. We add the sentences on result part (line 146-152, page 7, in the Methods) and (line 227-234 and 252-259 , page11and 12, in the results)
- d. Yes, we tested all probable risk factors that could be found in the database in the multivariate analysis. (line 155-162, page 8, in the Methods) and (line 225-232 and 254-261 page11and 12, in the results)
- e. We did not use Logrank and we delete the sentence.

3. Results

- a. In the description of patients, there is no information regarding the surgical technique and possible devices used.
- b. Please add annual revision rate and annual fusion rate with 95%CI
- c. Do you have the reasons of revision surgeries?

R: Thank you for the kind suggestion.

- a. We add the sentence regarding the surgical technique. (line 172-177, page 9 in the Results)
- b. We made the change of annual revision and fusion rate with 95%CI. (line 187-189and line 237-240 , page 9-11 in the Results)
- c. We calculated the result and add a section of " Reasons of Lumbar Spine Revision Surgery" and table S1.1 and table S1.2 (line 179-183, page 9, in the Results).

4. Discussion: OK.

R: Thank you. We are so grateful for all these valuable comments.

VERSION 2 – REVIEW

REVIEWER	Shahnaz KLOUCHE, MD Clinique du Sport Paris, FRANCE
REVIEW RETURNED	03-May-2018

GENERAL COMMENTS	Thank you for your revision. I recommend the acceptance of your manuscript.
---

REVIEWER	Hazem Elsebaie Cairo University, Egypt
REVIEW RETURNED	21-May-2018

GENERAL COMMENTS	The author's have addressed my concerns in this revision and I believe the study is acceptable for publication.
---

REVIEWER	G. SUDHIR
-----------------	-----------

	Sri Ramachandra Medical University, Chennai, Tamil Nadu , India
REVIEW RETURNED	25-May-2018
GENERAL COMMENTS	Suggestions have been accepted and corrected by the author